# Investigation on Response of Site of Typical Soil–Rock Composite Strata in Changchun Induced by Shield Construction of Parallel Twin Tunnels

**Liyun Li \*** and **Aijun Yao**

Key Laboratory of Urban Security and Disaster Engineering of Ministry of Education,
Beijing University of Technology, Beijing 100124, China; yaj@bjut.edu.cn
\* Correspondence: lly@bjut.edu.cn

**Abstract:** Underground engineering construction will inevitably change the stress state of surrounding strata, which will force a negative impact on the surrounding environment, even leading to the large deformation and damage of some adjacent structures. With a focus on the deformation of a typical soil–rock composite stratum site in the construction of Changchun Metro, relying on the shield construction of a parallel twin tunnel project between Northeast Normal University Station and Gong-Nong Square Station, which belongs to the Changchun Metro Line 1, the site deformation characteristics during the shield driving process of parallel twin tunnels were studied. Based on the data obtained from field monitoring and numerical simulation, ground settlement in shield driving was analyzed, the settlement trough was studied with the Peck formula, and the action of shield driving on the adjacent tunnel was discussed. Moreover, the influence range of shield driving was suggested, and the interaction between the twin tunnels with different axis spacings in shield driving was discussed. Some regular results obtained can provide support through data for similar projects in Changchun, China.

**Keywords:** soil–rock composite strata; shield tunneling; site deformation characteristics





## 1. Introduction

Underground engineering construction will inevitably change the stress state of surrounding strata, which will force a negative impact on the surrounding environment, even leading to the large deformation and damage of some adjacent structures. With the vigorous development of metro construction in China, safety issues in the construction of underground engineering have received widespread attention, and the deformation of strata during excavation is the most direct and significant issue.

Tunnels are an important component of urban subway systems. Many studies on site deformation induced by the excavation of tunnels were published. Peck [1] analyzed a large number of measured data on tunnel engineering and found that the ground settlement trough perpendicular to tunneling can be described by a normal distribution curve, i.e., a Gauss distribution curve. By assuming that the volume of ground settlement trough generated by tunneling approximately equals the amount of ground loss during excavation, he proposed a mathematical expression to describe ground settlement trough, known as the Peck formula, which has become a classical formula for predicting ground settlement trough in tunneling. Subsequently, many scholars [2–12] carried out a lot of research on the applicability of the Peck formula. Zhou et al. [13] developed a random forest (RF) model based on the Peck formula to predict ground settlements above tunnels, in which tunnel geometry, geological properties, and construction parameters were investigated as input variables to utilize in RF modeling. There are many other studies on ground settlement induced by the excavation of tunnels. For example, Verruijt and Booker [14] treated soil as a linear elastic material and provided an analytical solution for ground settlement caused

by tunnel excavation. Melis et al. [15,16] developed a numerical model to simulate the excavation procedure with earth pressure balance (EPB), taking into account the full excavation sequence. Lambrughi et al. [17] developed a three-dimensional numerical model for the simulation of the complete tunnel excavation process by EPB machines, and compared the influence of the linear elastic constitutive model, Mohr–Coulomb constitutive model, and modified Cam–Clay model on the calculation results. Fu et al. [18] developed an analytical approach to predict ground movement induced by a non-uniformly deforming circular tunnel in an elastic half-plane. Jiang et al. [19] investigated the deformation behavior of natural loess-experiencing complex stress paths around a shield tunnel. Ma et al. [20] studied the displacement characteristics of a "π"-shaped double cross-duct excavated via the cross-diaphragm (CRD) method. Zhou et al. [21] constructed the distribution density function of two random parameters, analyzed the sensitivity of parameters using the Monte Carlo method, and obtained the relationship curve of failure probability. Qiu et al. [22] suggested a simplified method for calculating the stresses and displacements of the surrounding rock and tunnel lining with time. Tao et al. [23] proposed an inverse method for improving the prediction of tunnel displacements during adjacent excavation. Deng et al. [24] proposed a hybrid regional model for predicting ground deformation induced by large-section tunnel excavation.

The works of Palmstrom and Stille [25] indicate that site deformation induced by tunnel excavation is controlled by the geological conditions of the site, the construction process, and the design scheme. With the rapid development of cities, the environment of subway construction is more and more complex, and the shield construction method has become an important means of tunneling. Therefore, it is very important to study the site deformation characteristics induced by shield tunneling under different ground conditions, explore the influence scope, and propose prevention measures.

Changchun Metro Line 1 is laid along Renmin Street, starting from North Huan-cheng Road Station in Kuan-cheng District and ending at Hong-zuizi Station in Nan-guan District. The stratigraphic structure in the Changchun region is a typical soil–rock composite stratum. The distribution of strata from top to bottom is fill, alluvial/diluvial clay, sandy soil, and mudstone. Studying the response of the site during the construction of the subway tunnel in the typical soil–rock composite strata of Changchun city is of great significance for accumulating experience for the implementation of subsequent metro projects. The authors of this paper conducted numerical works on this [26], discussed the mechanism of site deformation induced by shield tunneling, explored the influence of construction parameters on site deformation, and recommended that a reasonable value of earth chamber pressure in shield tunneling would be 1.0–1.5 times the static earth pressure.

This article is an extension of the author's previous work [26]. In this article, by sorting out the field monitoring data of the parallel twin tunnel project between Northeast Normal University Station and GongNong Square Station, which belongs to Changchun Metro Line 1, the characteristics of site deformation caused by shield tunneling were analyzed, and areas for strengthening monitoring were suggested. Meanwhile, based on the numerical models in the author's previous works [26], the interaction between the twin tubes in shield tunneling was discussed, with a view to gain some regular knowledge for the construction of Changchun Metro.

## 2. Project Overview

The subway tunnel is located below Renmin Street, starting from Ziyou Road and ending on Nanhu Road, as shown in Figure 1. From north to south, the subway tunnel passes on the side of many buildings, and passes through many rainwater, sewage, and thermal pipelines.

### 2.1. Engineering Geological Conditions

Figure 2 shows a partial geological profile along the tunnel, in which the location of the tunnel is also marked. Table 1 lists the lithological characteristics of strata.

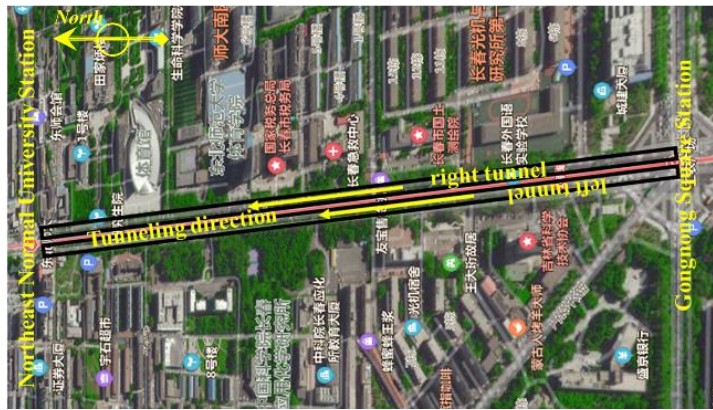

**Figure 1.** Location of this project.

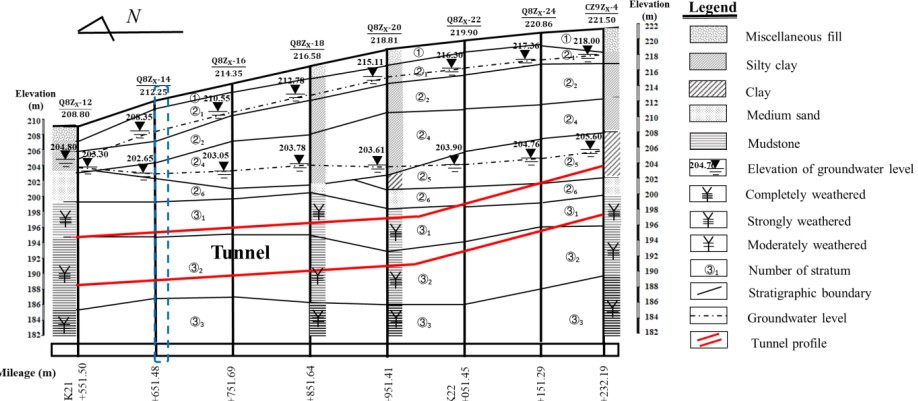

**Figure 2.** Geological longitudinal profile of the south section. The location circled by the blue dotted line is the area studied in this article.

## 2.2. Parameters of Tunneling

The topography of the project area is low in the north and high in the south, with the lowest elevation of 204.01 m and the highest elevation of 221.50 m. The subway tunnel is arranged in the form of parallel twin tubes, and the shape of the tubes is circular. The length of the tunnel is 1213.25 m, the outer diameter of the tubes is 6.0 m, and the axial spacing of twin tubes ranges from 14.3 m to 17.0 m. The maximum longitudinal slope of the tunnel is 25‰, and the buried depth of the tunnel ranges from 14.6 m to 22.5 m. According to the stratum conditions of the project area, an earth pressure balance (EPB) shield machine with a minimum turning radius of 250.0 m and a maximum slope of 5% was employed for tunneling in the project. The total length of the shield machine is 80.0 m, of which the length of the main machine is 9.0 m. The shield machine adopts the middle supported composite-steel cutter head made of Q345 + Q690, and the excavation diameter is 6280.0 mm. The double-edge hob, single-edge hob, tearing cutter, and soft rock cutter can be replaced to meet the needs of different geological conditions. The main drive system adopts an inner and outer double lip seal and a frequency conversion motor drive. The total length of the screw conveyor is 12.0 m, which is driven by hydraulic pressure. The belt conveyor is electrically driven and the conveying speed is 2.2 m/s. Synchronous grouting adopts single slurry grouting, and the injection pipeline type is a segmented detachable pipeline. A centrifugal pump cooling system is adopted; the flow rate of the pump is 900.0 L/min, and the pump lift is 70.0 m. The liquid capacity of the foam system is 750.0 L, and the foam is injected by a single-screw pump. The circular lining is composed of six prefabricated reinforced concrete segments assembled in a staggered manner, with a lining width of 1200.0 mm and a thickness of 300.0 mm. Table 2 lists the design parameters for tunnels, Table 3 presents the parameters of the shield machine, and Table 4 gives the initial parameters of shield tunneling.

**Table 1.** Lithological characteristics of strata.

| Stratigraphic Number | Geological Age | Lithology | Status | Thickness/m | Average Thickness/m | Steady Ability * | Grade of Surrounding Strata * | Distribution |
|---|---|---|---|---|---|---|---|---|
| ① | | artificial fill | slightly wet and slightly dense | 0.70–4.30 | 1.89 | Easy to collapse | VI | Whole |
| ②1 | Middle Pleistocene of Quaternary | silty clay | plastic to hard plastic | 0.90–4.30 | 2.41 | Bad | VI | Missing in some sections |
| ②2 | | silty clay | plastic or local soft plastic | 1.60–7.50 | 3.88 | Bad | VI | Whole |
| ②3 | | silty clay | plastic to hard plastic | 1.90–8.00 | 4.93 | Ordinary | VI | Found only in the south section |
| ②4 | | clay | plastic to hard plastic | 1.90–7.30 | 5.30 | Ordinary | VI | Found only in the south section |
| ②5 | | medium sand | medium-density to dense | 0.90–3.80 | 2.04 | Poor | VI | Found only in the south section |
| ③1 | Cretaceous | completely weathered mudstone | layering structure and original rock structure was destroyed | 3.10–9.60 | 5.82 | Poor | VI | Whole |
| ③2 | | strongly weathered mudstone | layering structure with developed joints and fissures | 5.60–15.00 | 8.50 | Ordinary | V | Whole |
| ③3 | | moderately weathered mudstone | layering structure with well-developed joints and fissures | 7.00–39.00 | - | Ordinary | V | Whole |

* The "stability ability" and "grade of surrounding strata" are determined in accordance with "Standard for engineering classification of rock mass" (GB/T 50218-2014) [27] and "Code for geotechnical investigations of urban rail transit" (GB 50307-2012) [28].

**Table 2.** Design parameters for tunnels.

| Items | Outer Diameter of Hole/mm | Inner Diameter of Hole/mm | Number of Segments per Ring | Segment Width/mm | Segment Thickness/mm | Design Strength Grade of Segment Concrete * | Impermeability Grade * |
|---|---|---|---|---|---|---|---|
| Value | 6000.0 | 5400.0 | 6 | 1200.0 | 300.0 | C50 | P10 |

* Taken according to "Standard for quality control of concrete" (GB 50164-2011) [29].

**Table 3.** Parameters of shield machine.

| Items | Length of Shield Body/m | Cutter Head Diameter/mm | External Diameter of Shield Shell/mm | Length of Shield Shell/mm |
|---|---|---|---|---|
| Value | 9.0 | 6280.0 | 6250.0 | 6000.0 |
| Items | Thickness of shield shell/mm | Maximum driving speed/(mm/min) | Maximum thrust/ton | Maximum working pressure/bar |
| Value | 40.0 | 100.0 | 3600.0 | 5.0 |
| Items | Maximum design pressure/bar | Cutterhead opening ratio | Excavation diameter/mm | Overbreak diameter/mm |
| Value | 6.0 | 35% | 6280.0 | 6364.0 |
| Items | Number of overbreak cutters | Number of wear detection knives | Blade spacing between scrapers/mm | Cutter spacing between hobs/mm |
| Value | 2 | 2 | 10.0 | 90.0 |
| Items | Axial turning force of hob/N·m | Number of mud and foam entrance | Rated torque of main drive system/ton·m | Lifting capacity of segment erector/W * |
| Value | 15.0–25.0 | 7 | 685.0 | 1.3 |
| Items | Total thrust of propulsion system/kN | Grease supply pressure of shield tail grease system/bar | Transport capacity of belt conveyor/(ton/h) | Slag output of screw conveyor/(m$^3$/h) |
| Value | 36,000.0 | 2050.0 | 800.0 | 430.0 |
| Items | Grouting capacity/(m$^3$/h) | Pumping capacity of foam system/(L/min) | Mixing capacity of bentonite system/(L/min) | |
| Value | 22.0 | 2000.0 | 500.0 | |

* Note: W means segment weight.

**Table 4.** Initial parameters of shield tunneling.

| Items | Thrust Force of the Jack/Ton | Driving Pressure/kPa | Grouting Pressure/kPa | Tunneling Velocity/(mm/min) |
|---|---|---|---|---|
| Value | 120.0 | 200.0 | 150.0 | 50.0 |
| Items | Earth chamber pressure/kPa | Slurry type | Synchronous grouting quantity/(m$^3$/m) | |
| Value | 160.0 | Single slurry | 5.0 | |

## 3. Research Methods

On-site monitoring is the most direct method with which to solve the issue of site response induced by tunneling. Therefore, ground settlement was monitored during the construction process. The monitoring points of longitudinal ground settlement and of transverse ground settlement were arranged along and perpendicular to the tunneling, respectively. The monitoring points of longitudinal ground settlement were located above the axis of the tubes. The monitoring sections of transverse ground settlement were set at the shield launching section, receiving section and connecting passages. To better understand the deformation characteristics of the site during shield tunneling, focusing on the study area, marked with a blue dashed box in Figure 2, three additional transverse monitoring

sections were specially positioned with an interval of 10.0m at K21 + 658, in which the buried depth of the tunnel axis was 22.5 m and the axis spacing of the twin tunnels was 14.4 m, as shown in Figure 3. According to the actual conditions of the area, 15 monitoring points were set up at each section, and the distance between neighbor monitoring points was 1.8 m to 7.7 m, as shown in Figure 4. The monitoring points of $A_1$, $B_1$, and $C_1$ were directly above the right tube, and the monitoring points of $A_2$, $B_2$, and $C_2$ were directly above the left tube. According to the existing literature [30] and to experience, the freezing depth in Changchun area is about 1.7 m below the ground. In order to reduce the effect of seasonal frost heaving, the rebar should be inserted below the frost line. Thus, the monitoring points for settlement were pins buried below the road surface, that is, as shown in Figure 5, a circular vertical hole with a diameter of 80 mm was drilled into the ground, of which the depth was more than 1.7 m; then, a reinforced bar was driven into the stable soil layer, and the depth of penetration was more than 200.0 mm to ensure the stability of the inserted reinforced bar. Finally, the circular vertical hole was backfilled with fine sand, and a protective sleeve and cover were set at the porthole.

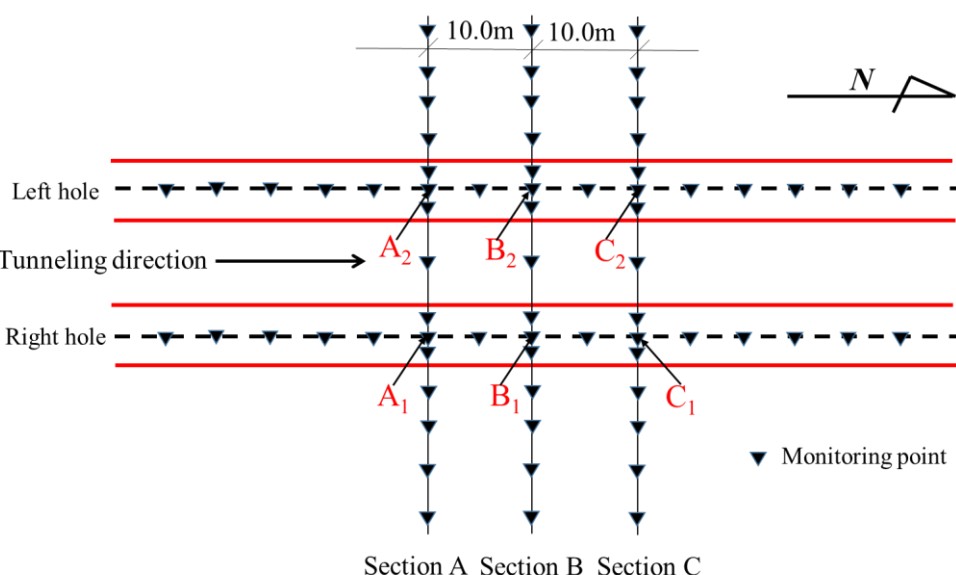

**Figure 3.** Schematic diagram of monitoring station for ground settlements.

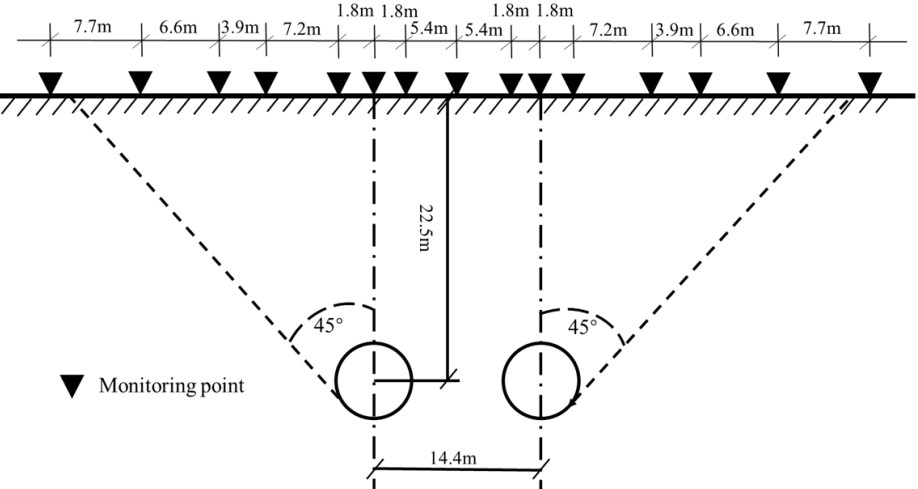

**Figure 4.** Location of points on transverse monitoring section.

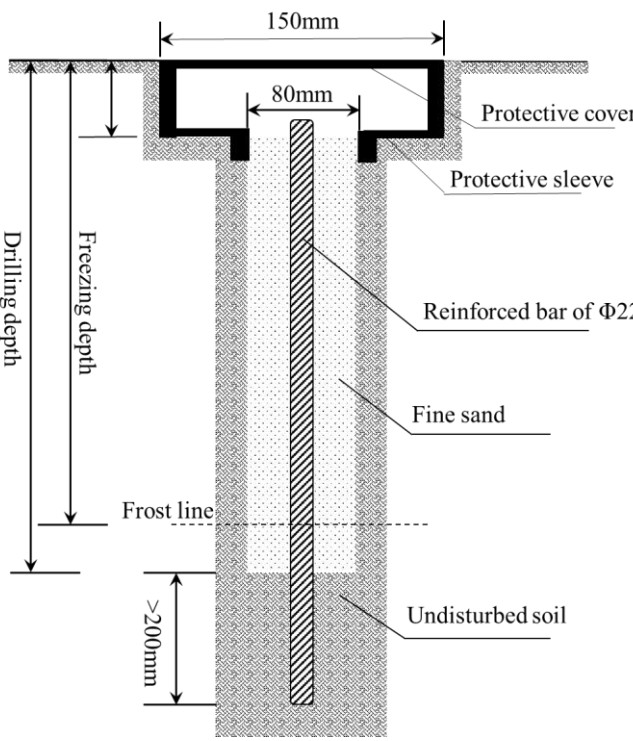

**Figure 5.** Schematic diagram of settlement point embedding.

The interaction between excavation and surrounding structures is an issue of concern [31]; thus, the lateral convergence and the heaving of the right tube were measured during the shield driving of the left one in order to understand the effect of shield tunneling on the adjacent tunnel.

Unfortunately, field monitoring for the displacement of deep strata and the internal force of tunnel segments was not carried out due to the limitation of construction conditions and a shortage of funds, which needed to be supplemented via numerical simulation. In this article, the FLAC$^{3D}$ software developed by Itasca Consulting Group, Inc. was employed to simulate the shield tunneling process. The numerical model was constructed based on monitoring section C. In numerical modeling, the thickness of the stratum and the physical–mechanical parameters of strata were taken according to those in the actual project, and the strata were assumed to be homogeneous, with a constant thickness and being horizontally layered. The two tubes were arranged in parallel, with a depth of 22.5 m on the tunnel axis and an axis spacing of 14.4 m between the two tubes. Tables 5 and 6 list the physical and mechanical parameters of strata and the mechanic parameters of structural members, respectively.

**Table 5.** Physical and mechanical parameters of strata [26].

| Name of Strata | Density/(kg/m$^3$) | Shear Modulus/MPa | Bulk Modulus/MPa | Poisson Ratio | Cohesion/kPa | Internal Friction Angle/(°) | Thickness/m |
|---|---|---|---|---|---|---|---|
| Fill ① | 1750 | 5.72 | 13.15 | 0.31 | 18.0 | 10.0 | 1.4 |
| Silty clay ②$_1$ | 1970 | 7.69 | 16.66 | 0.30 | 53.4 | 15.4 | 2.6 |
| Silty clay ②$_2$ | 1970 | 11.11 | 33.33 | 0.35 | 32.9 | 18.6 | 3.0 |
| Clay ②$_4$ | 1990 | 13.67 | 26.51 | 0.28 | 57.4 | 14.4 | 4.3 |
| Medium sand ②$_6$ | 2020 | 12.02 | 20.11 | 0.25 | 1.0 | 35.0 | 2.2 |
| Completely weathered mudstone ③$_1$ | 1940 | 19.35 | 30.76 | 0.24 | 32.1 | 21.1 | 6.0 |
| Strongly weathered mudstone ③$_2$ | 2010 | 24.39 | 37.03 | 0.23 | 60.0 | 20.0 | 6.5 |
| Moderately weathered mudstone ③$_3$ | 2310 | 49.58 | 68.96 | 0.21 | 70.0 | 18.8 | 26.5 |

**Table 6.** Mechanic parameters of structural members [26].

| Structural Members | Density/(kg/m³) | Elastic Modulus/GPa | Poisson Ratio | Conversion Coefficient | Clearance Value of Shield Tail/m |
|---|---|---|---|---|---|
| Segmental lining | 2500.0 | 35.0 | 0.2 | - | - |
| Shield shell | 2500.0 | 200.0 | 0.2 | - | - |
| Clearance element | | 1.0‰ Es | | - | - |
| Equivalent layer | | 0.22 | 0.2 | 1.0 | 0.14 |

## 4. Deformation Characteristics of Site

### 4.1. Ground Settlements along the Direction of Tunneling

Figure 6 shows the variation in ground settlement at monitoring points directly above the axis of the left tube and of the right tube during shield driving, in which, when the cutter head of shield machine does not reach the monitoring section, the distance from the working face to the monitoring section is taken as negative value; otherwise, the distance from the working face to the monitoring section is taken as positive value. D is the diameter of the cutter head of the shield machine, and D equals 6280.00 mm in this project.

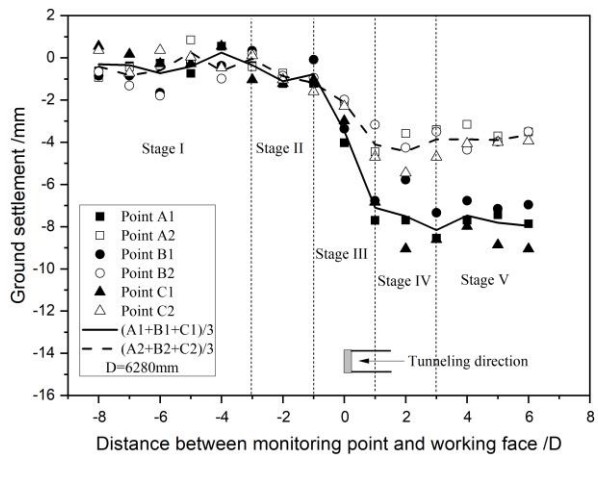

(**a**)

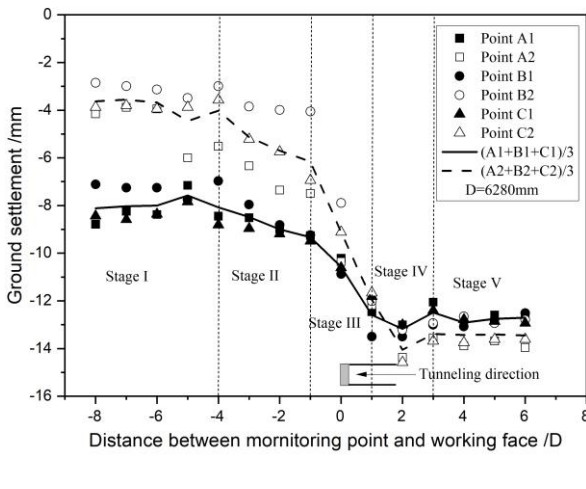

(**b**)

**Figure 6.** The settlement of the monitoring points following the movement of the working face. (**a**) The shield tunneling of the right tube; (**b**) the shield tunneling of the left tube.

The tunneling process of this project starts with the excavation of the left hole after completing the excavation of the right hole. Figure 6a gives the history curve of ground settlements during the shield tunneling of the right tube. It is found that the settlements of monitoring points $A_2$, $B_2$, and $C_2$ are smaller than those of the monitoring points $A_1$, $B_1$, and $C_1$, due to the disturbance induced by the excavation of the right tube to monitoring point $A_2$, $B_2$ and $C_2$ is smaller than to monitoring point $A_1$, $B_1$ and $C_1$. Because of the dynamic adjustment of shield tunneling parameters, the response of different monitoring points is slightly different, but the overall settlement characteristics are basically the same. From Figure 6a, the ground settlement at the monitoring points can be divided into five stages with the shield advance.

**Stage I. Small disturbance deformation**. When the monitoring section is more than 3D ahead of the working face, the deformation of the monitoring point is tiny. The ground may produce small settlement or uplift due to the loss of the stratum in front of the working face caused by the extrusion of the cutter head of the shield machine and excavation unloading.

**Stage II: Slow growth of deformation.** When the monitoring section is −1D to −3D away from the working face, the cutter head of the shield machine is close to the monitoring section, and the deformation of stratum at the monitoring section is greatly affected by the earth chamber pressure acting on the working face, which shows ground subsidence or

uplift. At this stage, stress release occurs in the stratum near the monitoring point. In order to ensure the stability of strata at the working face, the earth chamber pressure of the shield machine needs to be corrected in time.

**Stage III: Rapid growth of deformation**. When the working face is within 1D be-hind or ahead of the monitoring section, settlement increases and deformation grows rapidly. At this stage, when the cutter head of the shield machine passes through the monitoring section, there is a small gap between the shield shell and the surrounding strata, and sliding shear occurs at the interface between the shield fuselage and the surrounding strata; thus, the strata produce great disturbance. In this project, the ground settlement reaches 29% to 44% of the final settlement when the working face reaches the monitoring section position.

**Stage IV: Deformation increases abruptly again**. When the monitoring section is 1D to 3D behind the working face, the shield tail moves gradually away from the monitoring section. Because of the gap between the segmental lining and the surrounding strata, instantaneous ground subsidence occurs when the shield tail separates from the monitoring section. However, point $B_1$ presents upward uplift during this stage in this project due to a reduction in formation loss caused by an excessive grouting pressure and grouting volume. Therefore, a reasonable grouting pressure and grouting volume are needed in shield tunneling.

**Stage V: Deformation growth stops**. When the working face crosses the monitoring section by more than 3D, the shield machine is far away from the monitoring section, and the settlement tends to be a stable value.

Figure 6b describes the history curve of ground settlement during the shield tunneling of the left tube. As shown in Figure 6b, the initial settlement of each monitoring point is that induced by the former excavation of the right hole, and the settlement rate of monitoring points $A_2$, $B_2$, and $C_2$ above the left hole are faster than those of $A_1$, $B_1$, and $C_1$ during the shield tunneling of the left tube. After excavation, the final settlements of $A_2$, $B_2$, and $C_2$ are slightly larger than those of $A_1$, $B_1$, and $C_1$ because of the effect of segmental lining in the right tube. Similar to the shield tunneling of the right tube, ground settlement is also divided into five stages during the shield advance of the left tube. Due to the loss of the stratum near the left tube caused by the former excavation of the right tube, the disturbed areas of surrounding strata caused by the shield tunneling of the left tube extend a to 4D distance in front of the working face. The five stages of ground settlement during the shield tunneling of the left tube are $(-\infty, -4D)$, $[-4D, -D)$, $[-D, D]$ (D, 3D], and (3D, $\infty$), respectively.

As the above analysis, ground settlement gradually increases with the advance of working face during the former shield tunneling of the right tube. When the working face passes through the monitoring section and gradually moves away, the ground settlement tends to be a stable value. Subsequently, with the shield tunneling of the left tube, the ground settlement further increases, and finally, the ground settlement reaches the second stability. Due to the loss of stratum near the left tube caused by the former excavation of the right tube, the range of strata disturbance caused by the shield tunneling of the left tube extends to 4D in front of the working face. Therefore, for the shield construction of parallel twin tunnels in a Changchun soil–rock composite stratum, it is necessary to strengthen the monitoring of ground settlement in the range of 4D in front of the working face to 3D behind the working face.

*4.2. Ground Settlement Trough*

During shield tunneling, the ground settlement perpendicular to the tunnel axis tends to be the shape of a settlement trough with a large center and small sides. The method suggested by Peck [1] is widely used to predict ground settlement trough in tunnel construction, and its typical expression [3] is shown in Formulas (1)–(3):

$$S(x) = S_{max} exp \left[ \frac{-x^2}{2i^2} \right] \tag{1}$$

$$S_{max} = \frac{V_s}{\sqrt{2\pi i}} \quad (2)$$

$$V_s = \frac{V_l \pi D^2}{4} \quad (3)$$

in which S(x) presents the ground settlement value at the monitoring point, $x$ is the horizontal distance between the monitoring point and tunnel axis, $V_l$ is the volume loss rate of the stratum, and $i$ is the transverse profile inflexion distance.

Figure 7a shows the measured ground settlement of three monitoring sections caused by the tunneling of the right tube, as well as the corresponding curves fitted by Formula (1). As shown in Figure 7a, the ground settlements are symmetrical to the tunnel axis in the form of a normal distribution, which conforms to the description of Formula (1), except that the value of parameters in Formula (1) are different. At section A, the maximum ground settlement ($S_{max}$) is 7.95 mm, $i$ is 11.65 m, and $V_l$ is 0.75%. At section B, $S_{max}$ is 6.98 mm, $i$ is 10.89 m, and $V_l$ is 0.62%. At section C, $S_{max}$ is 9.01mm, $i$ is 10.48 m, and $V_l$ is 0.76%. In other words, $V_l$ in this project ranges from 0.62% to 0.76% during shield tunneling.

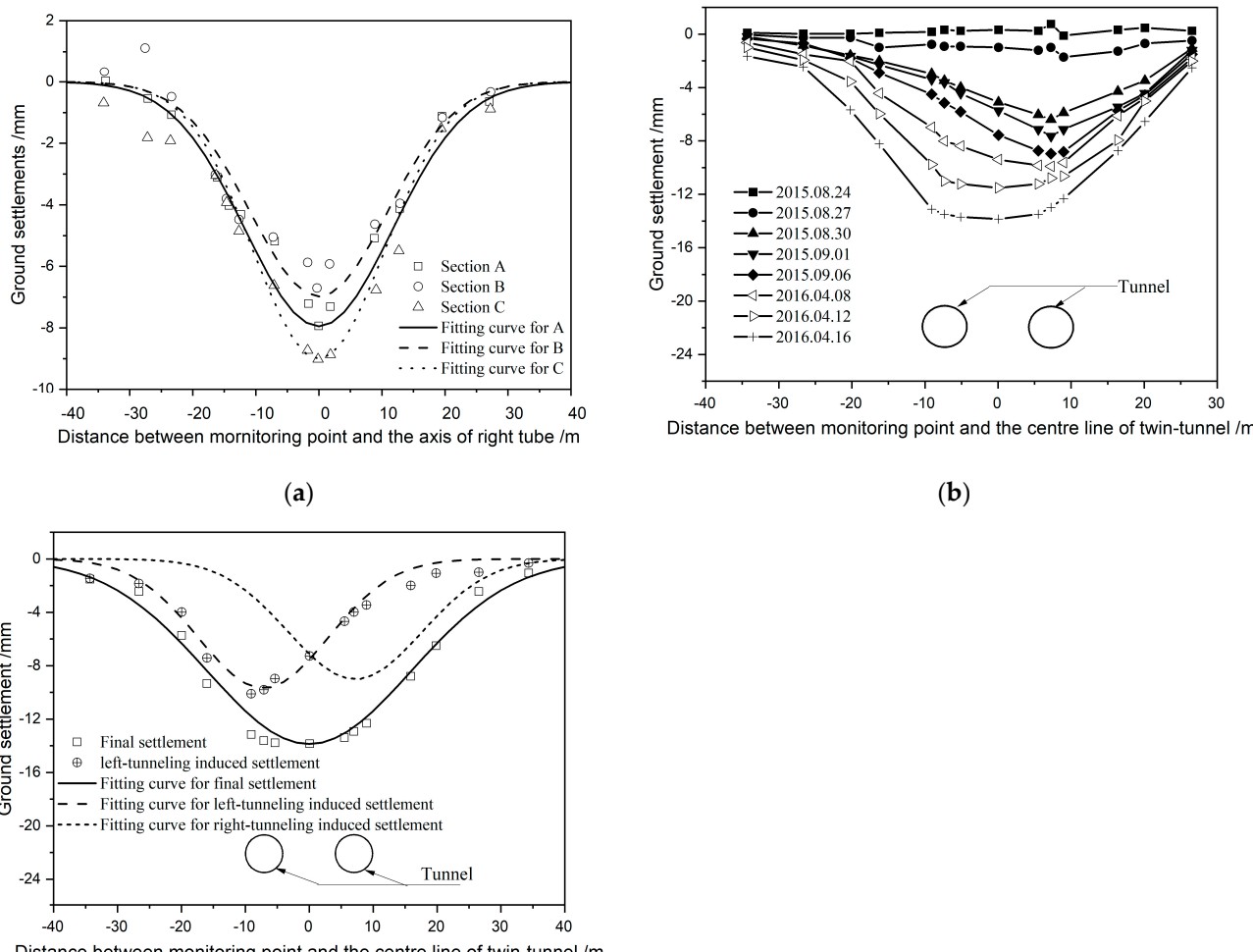

**Figure 7.** Ground settlement trough. (**a**) Ground settlement trough of the right tube; (**b**) Development of transverse ground settlement trough at section C; (**c**) final ground settlement curve at section C.

Figure 7b shows the development of ground settlement trough at section C during the construction of parallel twin tunnels. As shown in Figure 7b, ground settlement increases gradually with the former shield tunneling of the right tube; when the excavation of the right tube is completed, the ground settlement trough is symmetrical to the axis of the right

tube. Then, with the subsequent shield tunneling of the left tube, the ground settlement further increases, and the position of the maximum settlement moves to the left tube. After the completion of the double-tube construction, the ground settlement trough is approximately in a normal distribution, and the maximum ground settlement lies in the area between the two tubes.

Figure 7c gives the final ground settlement at section C and the fitting curve from Formula (1), which demonstrates that the ground settlement caused by the double tunneling is a U-shaped settlement trough. After excavation, the maximum ground settlement is 13.87 mm, and the transverse profile inflexion distance is 15.94 m. In order to reveal the disturbance of shield tunneling to the site, the ground settlement trough induced by the shield tunneling of the left tube ($S_{left}$) is also shown in Figure 7c. The value of $S_{left}$ is of the residual settlement after deducting the ground settlement induced by the former excavation of the right tube ($S_{right}$) from the total ground settlement ($S_{total}$), i.e., $S_{left} = S_{total} - S_{right}$. It can be seen that the maximum ground settlement induced by the shield tunneling of the left tube is 9.68 mm, which is larger than that induced by the former excavation of the right tube. The transverse profile inflexion distance caused by the shield tunneling of the left tube is 10.21 m, which is slightly smaller than that caused by the former excavation of the right tube. The volume loss rate of the stratum caused by the shield tunneling of the left tube is 0.80%, which is higher than that caused by the former excavation of the right tube. These results demonstrate that the former excavation of the right tube disturbs the surrounding strata, which leads to an increase in stratum loss and in ground subsidence during the shield tunneling of the left tube. In accordance with the field monitoring data, Table 7 lists the parameters in Formula (1).

**Table 7.** Parameter values of the Peck formula.

| Shield Driving | $S_{max}$/mm | $i$/m | $V_s$/m$^3$ | $V_l$/% |
|---|---|---|---|---|
| The right tunnel | 6.70–9.01 | 10.48–14.37 | 0.191–0.241 | 0.62–0.76 |
| The left tunnel | 9.68 | 10.21 | 0.248 | 0.80 |
| Double tunnels | 13.87 | 15.94 | 0.555 | 0.89 |

*4.3. Effect of the Shield Tunneling of the Left Tube on the Right One*

As mentioned above, the construction process of this project starts with the shield tunneling of the left tube after completing the excavation of the right tube. In order to study the effect of shield tunneling on the adjacent tunnel, the lateral convergence and the heaving of the right tube are monitored during the shield tunneling of the left one.

In shield tunneling, the driving pressure makes the surrounding strata float upward, but the unloading caused by the excavation of the tunnel creates a tendency of the surrounding strata to sink. Figure 8a shows the heaving of the tunnel floor of the right tube during the shield tunneling of the left tube, in which the positive value represents the upward floating of the tunnel floor and the negative value represents the sinking of the tunnel floor. As shown in Figure 8a, the heaving of the tunnel floor of the right tube can be characterized by four stages. When the working face is about 8D behind the monitoring point, the right tube begins to float up and down, but the amplitude is very small. When the working face is about 2.5D behind the monitoring point, the right tube uplifts rapidly. When the working face reaches 2.5D in front of the monitoring point, the uplift of the right tube begins to fall back. Finally, it reaches a stable value when the working face reaches 6D in front of the monitoring point. Due to the squeezing effect of synchronous grouting on soil, the final uplift of the right tube is about half of the maximum uplift.

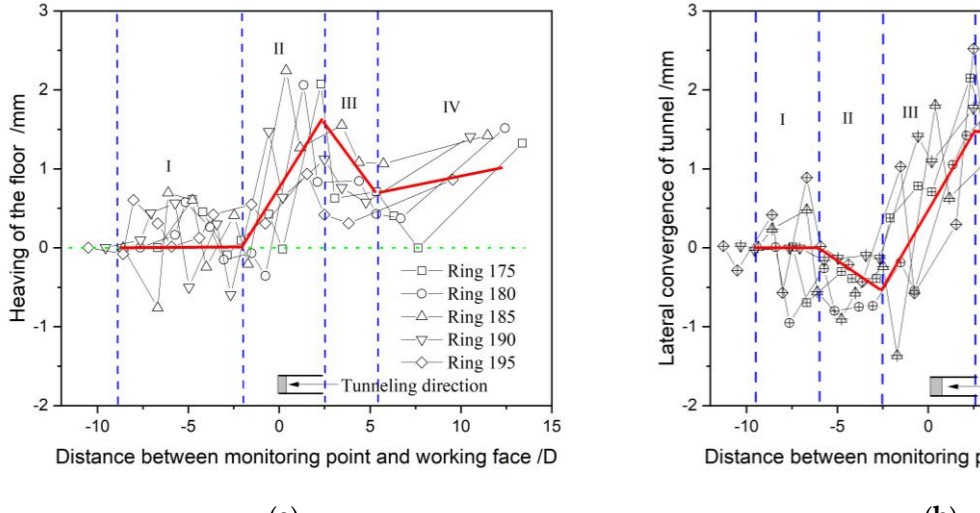

**Figure 8.** The response of the right tube under shield driving in the left tube. (**a**) Heaving of the floor; (**b**) lateral convergence. In figures, the red line shows the trend of changes in the heaving of the floor or lateral convergence.

Figure 8b presents the lateral convergence of the right tube during the shield tunneling of the left tube, in which the positive value indicates that the side wall of the tunnel pushes the surrounding strata (i.e., expansion), and the negative value indicates that the side wall of tunnel is separated from the surrounding rock (i.e., contraction). From Figure 8b, the lateral deformation of the right tube shows a tendency to first contract and then expand, and the value of deformation ranges from −1.0 mm to 3.0 mm. The lateral convergence of the right tube is also divided into four stages. When the working face is about 8D behind the monitoring point, the right tube begins to expand and contract in a fluctuating state. When the working face is about 6D behind the monitoring point, the right tube begins to contract. When the working face is about 2.5D behind the monitoring point, the right tube begins to expand. Finally, when the working face exceeds 2.5D in front of the monitoring point, the lateral convergence of the right tube reaches a stable value.

## 5. Analysis of Numerical Results

Due to the lack of on-site monitoring data on deep strata and the internal force of tunnel segments in shield tunneling in the project, a numerical simulation was employed as a supplement. The detailed numerical modeling process can be seen in another article [26].

Figure 9 shows the ground settlement on section C formed by shield tunneling. It can be seen that the distribution pattern of the settlement curve obtained from numerical simulation is basically consistent with the monitoring data, whether it is induced by the former right tunneling or by the twin tunneling. Due to the simplification of the geological strata in numerical simulation, there is a slight difference between the measured and numerical values, but the difference between them is less than 1.0 mm. This result proves that the numerical model is correct and reasonable.

Since we have analyzed the deep deformation characteristics of the site in reference [26], we only analyze the interaction between the twin tubes in shield tunneling in this section.

Figure 10 shows the mechanical responses of the right tube induced by the shield tunneling of the left tube, in which Figure 10a presents the horizontal displacement change in the side wall of the right tube with the advance of the working face, and Figure 10b gives the change in circumferential stresses of the right tube. It can be seen from Figure 10a that the right tube has a displacement pointing to the left tube being excavated. As the working face moves forward, the displacement difference between the left wall and the right wall of right tube, i.e., the lateral convergence, becomes more and more obvious, and cannot remain

stable until the working face is more than 2D in front of the monitoring point. Except for values smaller than those in the monitoring results, the change in lateral convergence of the tube is consistent with that in the field monitoring results shown in Figure 8b. As shown in Figure 10b, the circumferential stress at different positions of the right tube is different under the shield tunneling of the left tube. Taking point A as an example, when the working face reaches about 1D behind the monitoring point, the circumferential stress begins to increase, which is manifested as tensile stress with a maximum value of 30 kPa; when the working face reaches about 1D in front of the monitoring point, the circumferential stress begins to decrease, and when the working face reaches the front of the monitoring point at more than 1.5D, the circumferential stress maintains a stable value of about 20 kPa. The changes in circumferential stress at point B and point D are similar to that at point A except that they are forms of compressive stress. Point C is located at the bottom of the right tube, and the value of circumferential stress at the position fluctuates near zero with the advance of the working face.

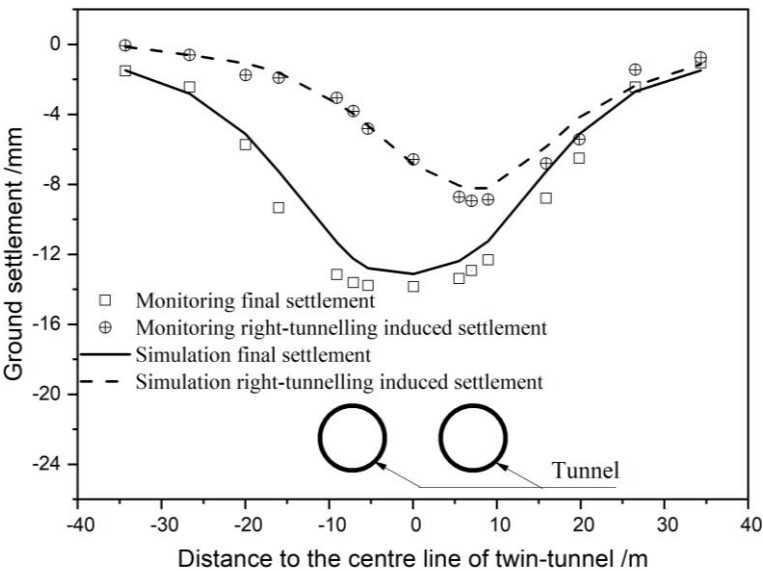

**Figure 9.** Model verification based on ground settlement on Section C.

Shield tunneling will lead to the inconsistent deformation of the adjacent tunnel. Therefore, the lateral deformation ratio (abbreviated as LDR) is defined, as shown in Formula (4), by normalizing the horizontal displacement difference at two monitoring points by using the distance between them.

$$\gamma_i = \frac{d_i - d_{i+1}}{L} \tag{4}$$

where $d_i$ and $d_{i+1}$ are the lateral displacements of point $i$ and of point $i + 1$ respectively. $L$ is the distance between the two monitoring points; the value is taken as the width of a single segment in this article.

Figure 11 shows the change in the LDR of the left wall of the right tube, which indicates that the LDR increases first, then decreases until it disappears with the advance of the working face, and decreases with the increase in the axis spacing of the twin tunnels. The peak value of the lateral deformation ratio (abbreviated as PLDR) slightly lags behind that of the working face, but the emergence of the PLDR will be synchronized with the working face when the axis spacing of the twin tunnels increases to a certain value. The two inflection points of the LDR curve appear at about −1D and 1.5D, respectively.

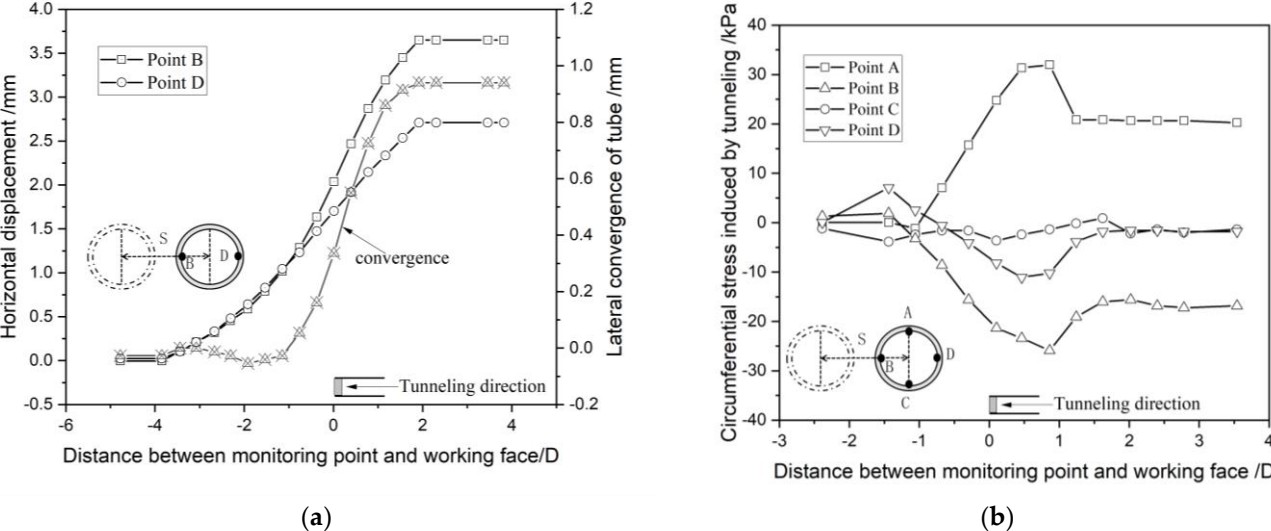

(**a**)                                                          (**b**)

**Figure 10.** Mechanical response of the first tube induced by the subsequent tunneling of the second tube. For displacement, a positive value means that the offset points to the tube being excavated, and a negative value means that the offset is far away from the tube being excavated. For convergence, a positive value indicates the expansion of the tube and a negative value indicates the contraction of the tube. For circumferential stress, a positive value represents tensile stress and a negative value represents compressive stress. (**a**) Horizontal displacement; (**b**) circumferential stress.

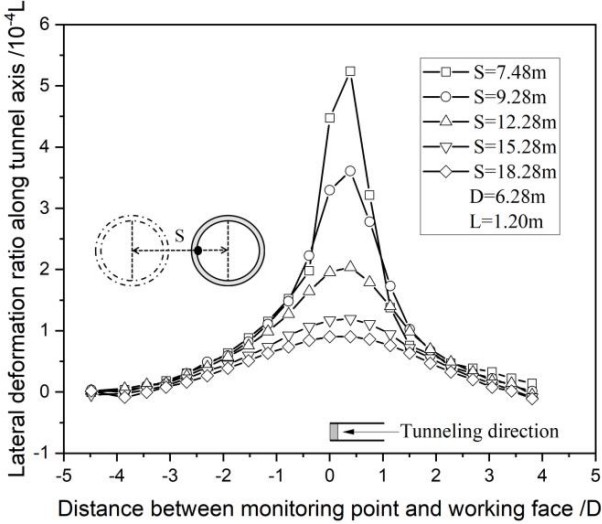

**Figure 11.** The lateral deformation of the first tunnel induced by shield driving in the second tunnel.

Figure 12 shows the duration curve of the circumferential stress at four different monitoring points of the right tube induced by the shield tunneling of the left tube. It is found that the circumferential stresses at the top and bottom of the right tube are tensile stress, while those at the left and right walls of the right tube are compressive stress, which means that the right tube causes the flattening deformation of vertical compression and horizontal expansion under left shield tunneling, and the cross-section of the right tube, initially a circle, becomes elliptical. The circumferential stress of the vault is greater than that of the arch bottom, and the circumferential stress of the left wall (close to the left tube being excavated) is greater than that of the right wall (far from the left tube being excavated). When the working face is about 1D behind the monitoring point, the circumferential stress of the right tube begins to increase until it reaches a peak value and quickly decreases to a residual value. The smaller the axis spacing of the twin tunnels, the greater the peak value and residual value of circumferential stress, and the later they appear.

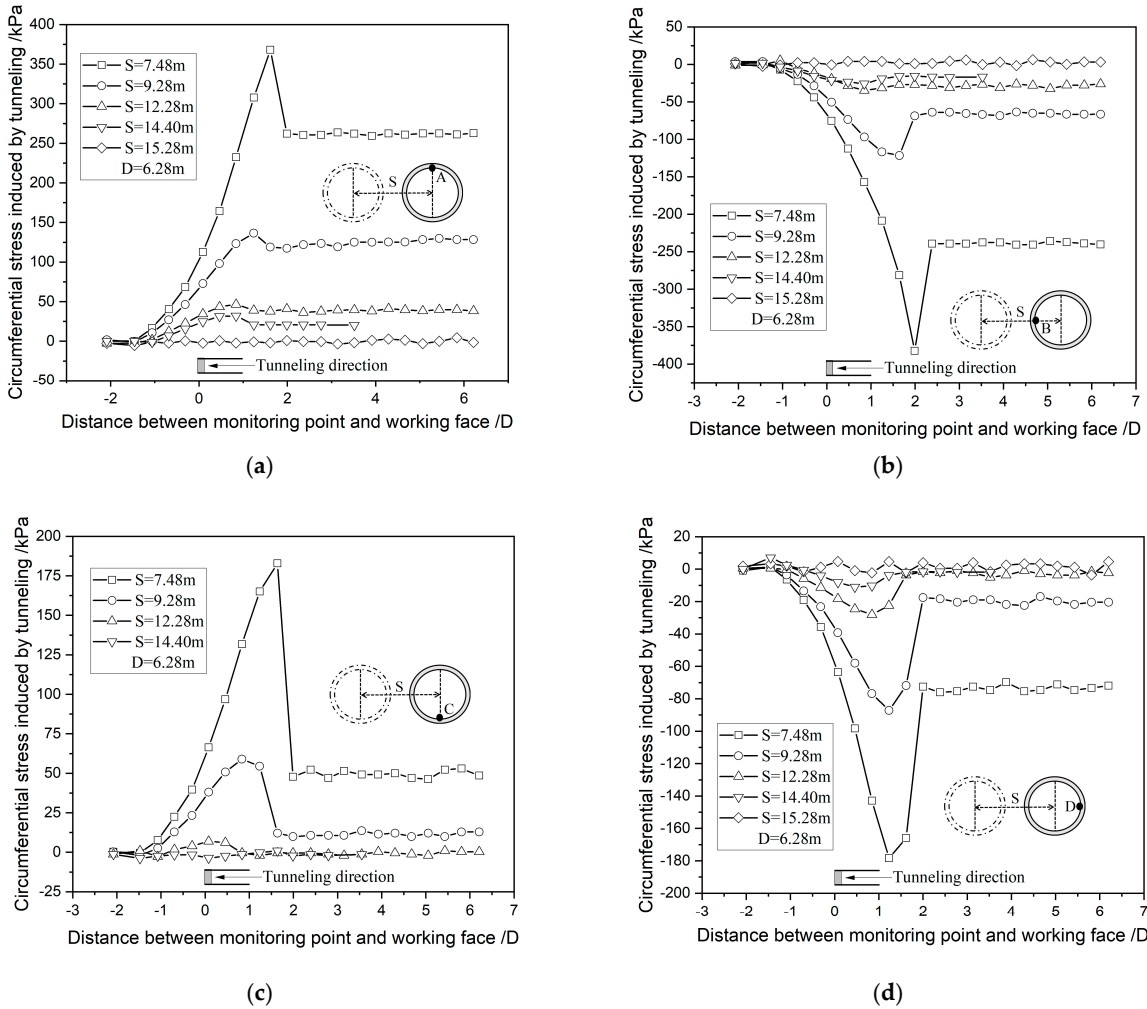

**Figure 12.** History of the circumferential stress of the first tunnel induced by the subsequent excavation of the second tunnel. (**a**) Point A; (**b**) point B; (**c**) point C; (**d**) point D.

Similarly, the existence of the adjacent tunnel also has a certain impact on the mechanical response of the tunnel being excavated. Table 8 shows the influence of the adjacent tunnel on the deformation of the tunnel being excavated. From Table 8, the existence of the adjacent tunnel reduces the displacement response of the tunnel being excavated, which means that the existence of the adjacent tunnel is beneficial to tunnel excavation. The influence on the arch crown of the tunnel is greater than that on the arch bottom, and the influence on the right wall is greater than that on the left side wall. With the increase in the axis spacing of the twin tunnels, the influence decreases monotonically.

**Table 8.** Deformation of shield segment in different cases (unit: mm).

| Cases | With an Adjacent Tunnel | | | | Without Adjacent Tunnel | | | | Influence of Adjacent Tunnel | | | | Relative Position between Two Tunnels |
|---|---|---|---|---|---|---|---|---|---|---|---|---|---|
| S/m | $Y_A$ | $X_B$ | $Y_C$ | $X_D$ | $Y_A$ | $X_B$ | $Y_C$ | $X_D$ | $Y_A$ | $X_B$ | $Y_C$ | $X_D$ | |
| 7.48 | −20.9 | −11.1 | 20.4 | 6.4 | | | | | −3.0 | −0.1 | −1.1 | −3.7 | |
| 9.28 | −22.0 | −11.2 | 20.6 | 7.8 | | | | | −1.9 | 0 | −0.9 | −2.3 | |
| 12.28 | −22.9 | −11.2 | 21.0 | 9.0 | −23.9 | −11.2 | 21.5 | 10.1 | −1.0 | 0 | −0.5 | −1.1 | |
| 15.28 | −23.1 | −11.2 | 21.4 | 9.6 | | | | | −0.8 | 0 | −0.1 | −0.5 | |
| 18.28 | −23.3 | −11.2 | 21.5 | 10.1 | | | | | −0.6 | 0 | 0 | 0 | |

Note: X and Y represent the horizontal displacement and the vertical displacement of the monitoring point, respectively; A, B, C, and D represent the monitoring points. The left tunnel is being excavated, and the right one is the adjacent existing tunnel.

## 6. Conclusions

In this article, we systematically investigated the site response caused by shield tunneling from Northeast Normal University Station to GongNong Square Station, which belongs to Changchun Metro Line 1. The response of the surrounding site during shield tunneling was analyzed; the deformation characteristics of the site and the interaction between the twin tubes in shield tunneling were discussed.

During shield tunneling, the ground settlement above the tunnel axis changes in an "*S*" shape, which can be roughly divided into five stages, i.e., small-disturbance deformation, slow growth of deformation, rapid growth of deformation, deformation increasing abruptly again, and deformation growth stopping.

During the shield driving of a single tunnel, the ground settlement trough shows a normal distribution, which can be fitted by the Peck formula. Within the portion of the tunnel studied in this article, the maximum ground settlement ranges from 6.7 mm to 9.01 mm, the transverse profile inflexion distance is 10.48 m to 14.37 m, and the volume loss rate of the stratum is 0.75% to 0.78%.

The lateral convergence of the right tunnel caused by shield driving in the left one is first slight expansion, then contraction, and finally expansion to a stable value. The LDR increases first and then decreases until it disappears with the advance of working face, and decreases with the increase in the axis spacing of the twin tunnels. The existence of the adjacent tunnel reduces the displacement response of the tunnel being excavated.

For the shield construction of parallel twin tunnels in the Changchun soil–rock composite stratum, it is necessary to strengthen the monitoring of ground settlement in the range of 4D in front of the tunnel working face to 3D behind the working face.

The numerical model can simulate the shield tunneling process well, and the ground settlement results are in good agreement. In terms of the response of adjacent tunnels, the calculated results show a similar trend to that of the measured results, except for the values that are smaller than those in the measured results. Therefore, a further in-depth numerical study is needed in the future.

**Author Contributions:** Conceptualization, L.L. and A.Y.; software, L.L.; validation, L.L. and A.Y.; investigation, A.Y. and L.L.; writing—original draft preparation, L.L.; writing—review and editing, L.L.; funding acquisition, L.L. All authors have read and agreed to the published version of the manuscript.

**Funding:** This research was funded by the Beijing Science and Technology planning Project, grant number Z181100009018001.

**Institutional Review Board Statement:** Not applicable.

**Informed Consent Statement:** Not applicable.

**Data Availability Statement:** All data, models, and codes generated or used during the study appear in the submitted article.

**Conflicts of Interest:** The authors declare no conflicts of interest.

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
