# Peer review of "Investigation on Response of Site of Typical Soil–Rock Composite Strata in Changchun Induced by Shield Construction of Parallel Twin Tunnels"

_applsci, doi:10.3390/app14020500_

Round 1
Reviewer 1 Report
Comments and Suggestions for Authors
A detailed investigation of the impact of shield tunneling from Northeast Normal University Station to Gong-Nong Square Station, which is a part of Changchun Metro Line 1 was conducted.
Based on the data obtained from field monitoring and numerical simulation, the ground settlement in the shield driving was analyzed, the settlement trough was studied with the Peck formula, and the action of shield driving on the adjacent tunnel was discussed. Moreover, the influence range of shield driving was suggested, and the interaction between the twin tunnels with different axis spacing in shield driving was discussed.
In the reviewer’s opinion, the paper can provide data support and may be useful support for similar projects.
Minor considerations
Line 109: The X vector should be defined before it is introduced. Please be sure to consider capital letters (or not) consistently.
Line 110: In the reviewer’s opinion, is a bit difficult to deduce relationships at Line 110 with reference to formulas presented at lines 277 and 400. The relationships should be deeply discussed and more specific.
Line 119: “Ac-cording” should be fixed.
Figures: in the reviewer’s opinion texts in figures should be bigger since they are difficult to read.
Author Response
Dear Reviewer,
Thank you for your comments!
I have carefully studied these comments for reviewing manuscripts, and revised the article in response to these suggestions. The revised content has been marked in red font in the revised version.
Now, I would like to reply to your comments.
A detailed investigation of the impact of shield tunneling from Northeast Normal University Station to Gong-Nong Square Station, which is a part of Changchun Metro Line 1 was conducted.
Based on the data obtained from field monitoring and numerical simulation, the ground settlement in the shield driving was analyzed, the settlement trough was studied with the Peck formula, and the action of shield driving on the adjacent tunnel was discussed. Moreover, the influence range of shield driving was suggested, and the interaction between the twin tunnels with different axis spacing in shield driving was discussed.
In the reviewer’s opinion, the paper can provide data support and may be useful support for similar projects.
Response:Thanks for your affirmation.
Minor considerations
Q1: Line 109: The X vector should be defined before it is introduced. Please be sure to consider capital letters (or not) consistently.
Response: Thanks for your suggestion. This descriptive text is from the template of this journal. Due to our operational error, we forgot to delete it during the typesetting of this article, and the paragraph has been deleted in the revised draft.
Q2: Line 110: In the reviewer’s opinion, is a bit difficult to deduce relationships at Line 110 with reference to formulas presented at lines 277 and 400. The relationships should be deeply discussed and more specific.
Response: Thanks for your suggestion. This descriptive text is from the template of this journal. Due to our operational error, we forgot to delete it during the typesetting of this article, and the paragraph has been deleted in the revised draft.
Q3: Line 119: “Ac-cording” should be fixed.
Response: Thank you for your suggestion. It was revised in the revision.
Q4: Figures: in the reviewer’s opinion texts in figures should be bigger since they are difficult to read.
Response: Thank you very much. We have enlarged the text in the Figures.
Reviewer 2 Report
Comments and Suggestions for Authors
Dear Authors.
I reviewed the proposed article titled: Investigation on the response of Site of Typical Soil-Rock Composite Strata in Changchun Induced by Shield Construction of Parallel Twin-tunnels
I think the article is interesting but needs corrections. The article presents an overview of the monitoring of pronounced deformations during the excavation of two tunnel tubes. From the point of view of transfer to engineering practise, the topic is welcome as it represents the establishment of monitoring and control of pronounced deformations that occur due to the relaxation of the material above the tunnel tubes during tunnelling. The topic covers the area of measurement and control of environmental influences during tunnelling.
I have the following comments on the content of the article:
1. I am not competent to evaluate English grammar and spelling, but I think it would be useful to check them.
2. References to some equations, tables and figures are several times distant and cause one to have to move through the text several times while reading, e.g. equation (1) mentioned in line 111 is actually in line 277, e.g. the reference to table 4 in line 136 is a few pages further on in line 147, e.g. the mention of figure 6 is found in line 207, and the figure itself is then a page further on in line 270. There are quite a few examples of this kind. This is quite exhausting for the reader and his concentration on the content.
3. I miss preliminary estimates of settlements/deformations calculated on the basis of theoretical principles and predictions in the article. Normally, predictions and calculations of environmental impacts are carried out before the start of project implementation. In a particular case of deformation. On this basis, monitoring of the area and measures to control deformations within the permissible limits are carried out.
4. In the discussion, I miss the comparison between the previously calculated data on the deformations and the measures to control the deformations within the calculated values.
5. In conclusion, I miss a comment on the deviation of the model from the actual situation and to the possibility of performing a back analysis based on data measured in situ.
Kind regards.
Author Response
Dear Reviewer,
Thank you for your comments!
I have carefully studied these comments for reviewing manuscripts, and revised the article in response to these suggestions. The revised content has been marked in red font in the revised version.
Now, I would like to reply to your comments.
I reviewed the proposed article titled: Investigation on the response of Site of Typical Soil-Rock Composite Strata in Changchun Induced by Shield Construction of Parallel Twin-tunnels
I think the article is interesting but needs corrections. The article presents an overview of the monitoring of pronounced deformations during the excavation of two tunnel tubes. From the point of view of transfer to engineering practice, the topic is welcome as it represents the establishment of monitoring and control of pronounced deformations that occur due to the relaxation of the material above the tunnel tubes during tunnelling. The topic covers the area of measurement and control of environmental influences during tunnelling.
Response:Thanks for your affirmation.
I have the following comments on the content of the article:
Q1. I am not competent to evaluate English grammar and spelling, but I think it would be useful to check them.
Response: Thank you very much! We have checked the manuscript again.
Q2. References to some equations, tables and figures are several times distant and cause one to have to move through the text several times while reading, e.g. equation (1) mentioned in line 111 is actually in line 277, e.g. the reference to table 4 in line 136 is a few pages further on in line 147, e.g. the mention of figure 6 is found in line 207, and the figure itself is then a page further on in line 270. There are quite a few examples of this kind. This is quite exhausting for the reader and his concentration on the content.
Response: Thanks for your suggestion! Due to our carelessness in work, there were input errors in the version you reviewed, which have been corrected in the revised draft.
Q3. I miss preliminary estimates of settlements/deformations calculated on the basis of theoretical principles and predictions in the article. Normally, predictions and calculations of environmental impacts are carried out before the start of project implementation. In a particular case of deformation. On this basis, monitoring of the area and measures to control deformations within the permissible limits are carried out.
Response: You are right. Indeed, a pre construction assessment is generally required before construction. However, these assessments are only conducted on the impact of construction on the surrounding environment and will not consider the impact during the construction process. The impact of the construction process should be given more attention.
Q4. In the discussion, I miss the comparison between the previously calculated data on the deformations and the measures to control the deformations within the calculated values.
Response: Thanks for your suggestion! The control of site deformation in shield tunneling construction is mostly achieved by adjusting the advancing speed, earth chamber pressure, and grouting pressure and volume behind the wall. In this article, we only discuss the deformation characteristics of the surrounding site induced by shield tunneling and the mutual influence of parallel tunnels, without discussing the control measures. The impact of construction parameters has been discussed in our another article published in KSCE Journal of Civil Engineering (DOI: 10.1007/s12205-020-0124-0).
Q5. In conclusion, I miss a comment on the deviation of the model from the actual situation and to the possibility of performing a back analysis based on data measured in situ.
Response: Thank you very much! The validation of the numerical model has been carried out in reference [31], and the numerical simulation in this article is based on reference [31].
Round 2
Reviewer 2 Report
Comments and Suggestions for Authors
Dear Authors.
I reviewed the proposed article titled: Investigation on the response of Site of Typical Soil-Rock Composite Strata in Changchun Induced by Shield Construction of Parallel Twin-tunnels
I think the article is interesting but needs corrections. The article presents an overview of the monitoring of pronounced deformations during the excavation of two tunnel tubes. From the point of view of transfer to engineering practise, the topic is welcome as it represents the establishment of monitoring and control of pronounced deformations that occur due to the relaxation of the material above the tunnel tubes during tunnelling. The topic covers the area of measurement and control of environmental influences during tunnelling.
In the preliminary review, I pointed out to the authors that they should add the results of the previous preliminary impact assessment and a discussion of the results between those previously identified by the modelling and those actually expressed in order to increase the reader's interest in the topic. Points 3, 4 and 5 of the reviewers' comments.
In re-reading the article and reviewing the authors' responses, I note that this type of analysis is not included in the article, or that there is a reference [31] to previous analyses, and that discussions of the proposed comparison are not included.
Insofar as we would like to see a substantive article showing the whole process from the preliminary assessment of the impact of excavation on both the surface and the adjacent tube and the results of monitoring the impact during implementation, we now have only about half of the whole subject before us, which is otherwise interesting.
In view of the fact that the authors were not satisfied with the presentation of the entire topic, I consider the article in its present form to be unsatisfactory. If the authors complete the article according to comments 3, 4 and 5, I am willing to re-evaluate the content. Otherwise, I think that the article is not detailed enough and does not present the whole process from planning to implementation to be interesting for the reader. I personally think the reference to [31], which I believe is from one of the authors, is inappropriate. The monitoring of tunnelling is a mandatory practise, and the measurement and interpretation of the results is necessary for the ongoing planning of safe works with the least possible impact on the environment. From this point of view, the article does not represent an innovative approach, but rather reflects standard procedures and measures in the context of underground works.
Kind regards.
Author Response
Thank you to the reviewer for acknowledging the topic of the article.
Perhaps we misunderstood the comments provided by the reviewer, so our first response did not meet the reviewer's requirements.
On the basis of understanding the comments of the reviewers, we checked the manuscript and found that there are indeed some issues. Therefore, we sorted and revised it, and highlighted the revised content in red font.
Regarding the Points 3, 4 and 5 of the reviewers' comments, we would like to re-reply as follows.
Q3. I miss preliminary estimates of settlements/deformations calculated on the basis of theoretical principles and predictions in the article. Normally, predictions and calculations of environmental impacts are carried out before the start of project implementation. In a particular case of deformation. On this basis, monitoring of the area and measures to control deformations within the permissible limits are carried out.
Response: You are right. I also believe that the prediction and calculation of environmental impacts have been carried out before the start of project implementation. But because these tasks were not done by us, we have not seen any relevant data. We are only conducting some further research based on the actual project. This article is an extension of the author's previous work, we have explained this and provided a brief introduction to the previous work in the introduction section of the revised draft.
Q4. In the discussion, I miss the comparison between the previously calculated data on the deformations and the measures to control the deformations within the calculated values.
Response: Thanks for your suggestion. In the project, the deformation was only controlled by dynamically adjusting the construction parameters of the shield machine. Due to the lack of detailed information on the shield tunneling construction parameters, we did not delve deeper into it and only suggested the reasonable value of earth chamber pressure during the construction process. I hope to have the opportunity to carry out more in-depth works in the future. In the revised manuscript, we selected the monitoring data of section C as the benchmark to validate the numerical model.
Q5. In conclusion, I miss a comment on the deviation of the model from the actual situation and to the possibility of performing a back analysis based on data measured in situ.
Response: Thank you very much! We have added this content in the revised manuscript.
Round 3
Reviewer 2 Report
Comments and Suggestions for Authors
Accepted.
Kind regards.